# A versatile nuclei extraction protocol for single nucleus sequencing in non-model species– Optimization in various Atlantic salmon tissues

**Rose Ruiz Daniels**[1]*, **Richard S. Taylor**[1], **Ross Dobie**[2], **Sarah Salisbury**[1], **James J. Furniss**[1], **Emily Clark**[1], **Daniel J. Macqueen**[1], **Diego Robledo**[1]

1 The Roslin Institute and Royal (Dick) School of Veterinary Studies, The University of Edinburgh, Edinburgh, United Kingdom, 2 Centre for Inflammation Research, The Queen's Medical Research Institute, Edinburgh BioQuarter, University of Edinburgh, Edinburgh, United Kingdom

* rose.daniels@roslin.ed.ac.uk

**Data Availability Statement:** The raw sequencing files are available from the Gene Expression

## Abstract

The use of single cell sequencing technologies has exploded over recent years, and is now commonly used in many non-model species. Sequencing nuclei instead of whole cells has become increasingly popular, as it does not require the processing of samples immediately after collection. Here we present a highly effective nucleus isolation protocol that outperforms previously available method in challenging samples in a non-model specie. This protocol can be successfully applied to extract nuclei from a variety of tissues and species.

## Introduction and background

Single cell RNA sequencing has become a standard tool for profiling transcriptomic diversity across thousands of individual cells, in a wide range of species and tissues [1]. A major limitation of this technology is that it typically requires the isolation and immediate processing of live cells from fresh tissue, which in many circumstances is not practical. As a result, single nucleus RNA sequencing (snRNA-seq), requiring the isolation of nuclei instead of the whole cells, has been widely adopted to allow the use of frozen samples that can be stored for several months prior to processing, while yielding comparable results [2, 3]. The most critical step in a snRNA-seq protocol is the successful isolation of high quality nuclei, with a choice several options available for the dissociation and nuclei isolation [4–6]. Extensive work has been performed in mammals to develop a toolbox of protocols for nuclei extraction in a variety of tissues [3–5]. A review of the available protocols, including consideration of their performance in non-model species, concluded that mincing the tissue in salt tween (TST) represents the most effective method for nuclear isolation from frozen tissue in terms of the diversity of cell types captured and reducing background noise [7].

Here we present an improved, more robust protocol for the extraction of high quality nuclei from a variety of tissues in non-model species. This protocol has been adapted from a TST based method previously considered the gold standard, which has been used previously in a

Omnibus (https://www.ncbi.nlm.nih.gov/geo/). Accession number GSE231945; Title: A versatile nuclei extraction protocol for single nucleus sequencing in non-model species - optimization in various Atlantic salmon tissues.

**Funding:** The author(s) received no specific funding for this work.

**Competing interests:** The authors have declared that no competing interests exist.

snRNA-seq study in Atlantic salmon liver [8], and which now results in superior data quality in a variety of tissues tested in Atlantic salmon. We apply the protocol to flash frozen skin samples from Atlantic salmon, which are challenging tissues to work with due to their toughness, the presence of connective tissue and fat deposits, and hard tissue such as scales. We include notes throughout the protocol to allow the user to optimise for a different tissue types. While the Chromium 10x platform was used to partition nuclei and prepare the snRNA-seq libraries, the nuclei are suitable for other platforms and applications. The aim of this protocol is to capture 7,000 nuclei per single-nuclei RNA sequencing library using the Chromium Single Cell 3' Reagent Kits v2 or v3 (10x Genomics). Given its utility for isolating nuclei from difficult to dissociate tissue types, we anticipate this protocol will be broadly applicable for snRNA-seq of non-model organisms and unconventional tissue types.

## Materials

### Sampling and storage for nuclear isolation

Approximately ~45mg (about the size of a grain of rice) of tissue is placed in a clearly labelled cryotube and immediately flash frozen in liquid nitrogen. It is critical that the tissue is preserved as rapidly as possible to minimise RNA degradation and cell stress that may induce transcriptional response. In the absence of liquid nitrogen, samples can be frozen in dry ice [8]. Samples should be stored at -80°C or colder, and should be processed as soon as possible, ideally within 3 months of flash freezing. Longer periods of storage are possible but nuclei and RNA quality will be reduced, impacting the final libraries.

### Reagents

Quantities listed here result in sufficient buffer to process two samples.

Notes on use of RNase inhibitor:

- The choice of RNase inhibitor can negatively affect RNA integrity during nuclear isolation and all steps prior to reverse transcription. The RNase inhibitor used here was recommended by 10x for use with their platforms, providing superior performance to alternatives[8]. For alternative platforms, consult with the protocol or manufacturer for a compatible product.

- Add RNase inhibitor to buffers immediately prior to nuclear extraction.

- RNase inhibitor is only necessary for samples destined for downstream sequencing. For trials that will not be sequenced, the RNase inhibitor can be omitted to save on costs.

- RNase inhibitor concentration can be increased up to 1000U/ml depending on RNase activity levels in the tissue. Some downstream applications, such as multi-omics ATAC-seq/RNA-seq, require higher levels of RNase inhibitor.

### Workflow

Pre-chill the centrifuge to 4°C. Samples should be kept frozen on dry ice or liquid nitrogen until immediately prior to nuclear isolation. All subsequent sample-handling steps should be performed on ice. Make sure all materials are available Table 1 before starting protocol. All buffers should be chilled on ice. Make sure buffers are available Table 2, all concentrations are adjustable to number of samples required. Start by preparing 2x ST (Table 3) as master buffer to prepare the rest. Subsequently make ST (Table 4), TST (Table 5) and PBS+0.02% BSA (Table 6) prior to starting the isolation,

**Table 1. List of materials required for the correct implementation of this protocol, as well as the supplier and catalogue number.**

| Material | Supplier | Catalogue number |
|---|---|---|
| Noyes Spring Scissors | Fine Science Tools | 15514–12 |
| Tungsten Carbide Straight 11.5 cm Fine Scissors | Fine Science Tools | 14558–11 |
| 40 μm Falcon™ cell strainer | Thermo Fisher Scientific | 08-771-2 |
| 30 μm Falcon™ cell strainer | Corning | 352235 |
| 20 μm cell strainer for 1.5ml tubes | pluriSelect | 43-10020-50 |
| LoBind Tubes 1.5 ml | Eppendorf | 0030108051 |
| 6 well Tissue Culture plate | Stem Cell Technologies | 38016 |
| Falcon tubes 15 ml | corning | CLS430055 |
| C-chip disposable haemocytometer | VWR | 82030–468 |

**Table 2. Complete list of reagents and volumes required to make up the buffers for the correct implementation of this protocol.**

| Reagent | Supplier | Product code | Volume (μl) |
|---|---|---|---|
| NaCl | Thermo Fisher Scientific. | AM9759 | 292 |
| Tris-HCl pH 7.5 | Thermo Fisher Scientific | 15567027 | 100 |
| $CaCl_2$ | VWR | E506-100ml | 10 |
| $MgCl_2$ | Sigma–Aldrich | M1028 | 210 |
| Nuclease-free water | VWR | E476-500ml | 14245 |
| Protector RNase inhibitor | Sigma Aldrich | PN-3335399001 | 31 |
| 1% Tween-20 | Sigma Aldrich. | P-7949 | 120 |
| 2% BSA | New England Biolabs. | B9000S | 20 |
| Molecular grade PBS | | | 985 |

**Table 3. Reagents and amounts to prepare 2x salt-Tris (ST) (10 ml).**

| Reagent | Volume (μl) | Final concentration (mM) |
|---|---|---|
| NaCl | 292 | 146 |
| Tris-HCl pH 7.5 | 100 | 10 |
| $CaCl_2$ | 10 | 1 |
| $MgCl_2$ | 210 | 21 |
| Nuclease-free water | 9,388 | |

**Notes**: Prepare fresh 2xST on the day for each isolation. Chill prior to use.

**Table 4. Reagents and amounts to prepare 1x ST with RNase inhibitor (6ml).**

| Reagent | Volume (μl) | Final concentration |
|---|---|---|
| 2xST | 2,997 | |
| Nuclease-free water | 2,997 | |
| Protector RNase inhibitor | 6 | 40U/ml |

**Notes**: Add RNase inhibitor immediately prior to use. RNase inhibitor volume can be increased up to 1000U/ml if the target tissue is rich in RNases (e.g. spleen; pancreas).

**Table 5. Reagents and amounts to prepare TST with RNase inhibitor (4ml).**

| Reagent | Volume (µl) | Final concentration |
|---|---|---|
| 2xST | 2,000 | |
| 1% Tween-20 | 120 | 0.03% |
| 2% BSA | 20 | 0.02% |
| Nuclease-free water | 1,840 | |
| Protector RNase inhibitor | 20 | 200U/ml |

**Notes**: Prepare fresh TST on the day of the isolation. Chill prior to use. Prepare the 1% Tween from the 10% Tween stock solution using nuclease free $H_2O$. RNase inhibitor volume can be increased up to 1000U/ml if the target tissue is rich in RNases.

**Table 6. Reagents and amounts to prepare PBS+0.02% BSA with RNase inhibitor (1ml).**

| Reagent | Volume (µl) | Final concentration |
|---|---|---|
| Ultra-pure molecular grade PBS | 985 | |
| 2% BSA | 10 | 0.02% |
| Protector RNase inhibitor | 5 | 200U/ml |

**Notes**: It is possible that nuclei may stick together in clumps after centrifugation. This can be observed under a microscope (Fig 1A illustrates an example of this). This is undesirable as it may lead to high rates of "doublets", when more than one nucleus is encapsulated in a single droplet in the microfluidics system, resulting in each transcript being labelled with the same cellular barcode. To prevent this, BSA concentration can be increased up to 2%. This protocol assumes the nuclei will be suspended in a final buffer of 500 µl of PBS+BSA, as recommended for the 10x Chromium system. This final buffer may vary for alternative platforms and tissues adjust volumes accordingly.

1. Place a 6-well tissue culture plate on ice and add 1ml of TST to one well. Place the frozen tissue sample into the well containing the TST. If the sample is stuck to the cryotube, remove it using tweezers while ensuring the sample does not defrost, and place immediately into the culture plate with TST.

2. Keeping the culture plate on ice, mince tissue initially using Tungsten Carbide scissors for 30 seconds (if required) and then with Noyes Spring Scissors (Fine Science Tools, catalog no. 15514–12) for up to 5 minutes until finely minced. Using a P1000 with a low retention filter tip, gently pipette up and down for up to a further 5 minutes. The total time in the dissociation buffer is critical and the duration of the scissor and pipette steps needs optimisation using non-valuable samples. The use of scissors should be the minimum time necessary for no solid lumps of tissue to remain. Pipetting duration is optimised by observing the nuclei under a microscope and adjusting the dissociation time to the minimum needed for full dissociation and lysing of cells (Fig 1). Typical tissue dissociation takes a total of 5–10 minutes. Non-tissue samples such as blood samples require as little as 1 minute of gentle pipetting in the TST and no use of scissors.

3. Pass lysate through a 40µm strainer into an empty well in the tissue culture plate and wash the cell strainer with 1ml TST. Add 3ml of chilled 1xST buffer to the lysate to stop the reaction. Move the 5ml of lysate to a labelled 15ml falcon tube on ice.

4. Centrifuge at 4°C for 5 minutes at 500g in a swinging bucket centrifuge. When nuclei yield is low, centrifugation time can be increased to 10 minutes to maximise yield. In samples with high recovery 10 minutes is not recommended as longer centrifugation duration can result in the clumping of nuclei and higher doublet rates in the final data.

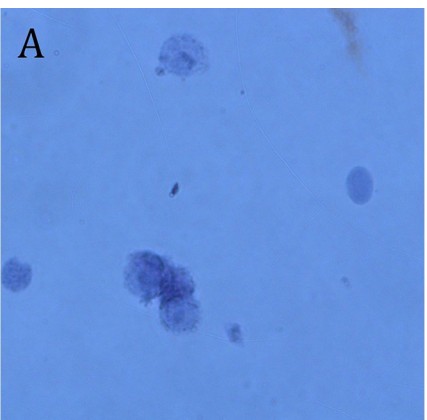
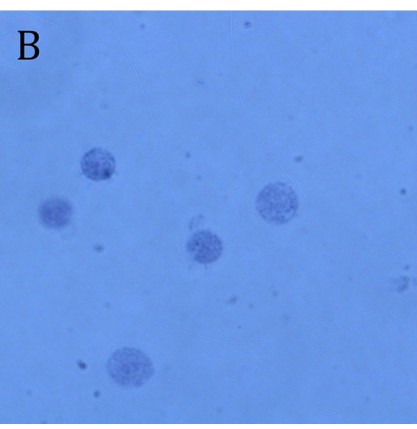
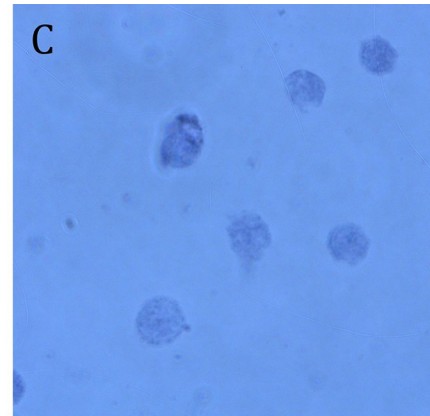

**Fig 1. Three examples of nuclear dissociation trials in Atlantic salmon liver at x40 magnification, stained with trypan blue. A**. Nuclei did not spend sufficient time in the dissociation buffer resulting in incomplete dissociation. This may result in the obstruction of the microfluidic device and failure of the library. **B**. Nuclei perfectly dissociated showing minimal aggregation, ideal for library preparation. **C**. Nuclei have been in the dissociation buffer for too long, resulting in degradation of the nuclear membrane, which will result in suboptimal library quality.

5. Discard the supernatant with a P1000 pipette, and gently resuspend the pellet in PBS-BSA. Resuspension volume depends on the size of the pellet and recommended nucleus concentration levels for the downstream platform. We recommend 1ml for most tissues and 100–500μl for more difficult tissues that yield fewer nuclei. For skin and fin, 400μl is recommended.

6. Filter the nucleus solution a second time. The size of the filter will depend on the type of tissue, e.g. for tissues such as liver and head kidney a 40μm cell strainer will suffice, whereas for gill a 30μm filter may be better given the higher amount of tough debris, which could negatively impact later stages. For tissues that produce a lot of debris such as fin and skin, 20μm is recommended. If the lysate does not pass through at once, pipette up and down very gently with a wide bore pipette.

7. A) Count the nuclei using a haemocytometer. In this step, the nuclei may also be examined under a microscope to ascertain the level of debris present and the integrity of the nuclear membranes. High levels of debris may indicate that a finer filter should be used, or using less tissue initially. Incomplete dissociation or evidence of damaged nuclear membranes indicates that the time spent in the TST should be adjusted (Fig 1).

   B) The nuclei may also be counted using a Bio-Rad TC20, or similar platform, to count the proportion of viable cells. Nuclei are identified as "dead", therefore a good nuclei isolation will have a small percentage of live cells. <4% live cells is ideal, but <12% is acceptable. A high proportions of live cells indicates incomplete nuclear isolation, and more time in the TST solution is needed. We advise against trusting the nucleus count given by automated counters, as they often substantially underestimate the true number of nuclei present. This would result in excessive nuclei being loaded into the Chromium Controller, leading to high doublet rates and low sequencing saturation.

8. Load the nuclei suspension into a Chromium Chip and into the Chromium Controller, aiming to recover 7,000 nuclei as per 10x recommendations with a concentration of between 700 to 1200 nuclei per μl. In the case of some tissues such as fin, re-adjust the target recovery to 5000 as the concentration of nuclei obtained can be lower.

## Testing the protocol

Using 45mg flash frozen Atlantic skin samples, we compared a version of the mincing+ST nuclear isolation protocol (V1), to our new protocol (V2), where we optimized several steps to improve performance. The loading buffer was changed from ST to PBS+BSA and RNAse inhibitor, RNA inhibitor was also added to all buffers, the mincing step was adjusted by using a combination of Noyes spring scissors, Carbide scissors and pipetting up and down. Filter size was decreased to remove more debris, and the centrifugation length optimized to enhance nuclei recovery.

After isolation with the two different protocols, nuclei were processed with the 10x Chromium™ Single Cell Platform using the Chromium™ Single Cell 3'Library and Gel Bead Kit v3.1 and Chromium™ Single Cell A Chip Kit (both 10x Genomics) as per the manufacturer's protocol, with a target recovery of 7000 nuclei per sample. The nuclei were loaded into a channel of a Chromium 3' Chip and partitioned into droplets using the Chromium controller before the captured RNA for each nucleus was barcoded and reverse transcribed. The resulting cDNA was PCR amplified for 14 cycles, fragmented, and size selected before Illumina sequencing adaptor and sample indexes were attached. Libraries were sequenced on a NovaSeq 6000 by Novogene UK Ltd (2x150bp paired end reads).

Raw sequencing data were aligned to the unmasked ICSASG_v2 reference assembly (Ensembl release 104) of the Atlantic salmon genome. The analysis used just protein coding genes. Mapping of reads to the genome, assignment of reads to cellular barcodes, and collapsing of unique molecular identifiers (UMIs) was performed using STARsolo v2.7.7a [9]. Settings as described in [8]. The top 100,000 cell barcodes ranked by UMI number were retained to ensure the capture of transcriptionally quiet nuclei, these are normally lost when using the automated STARSolo filtering algorithm. Mapping statistics for each snRNA-Seq sample are provided in Table 7.

## Results and discussion

Flash frozen Atlantic salmon skin was used to test both TST extraction protocols. Skin is a challenging tissue on which to perform nuclear isolations. Thus, our rationale was that this protocol will transfer well to a range of tissues with difficult properties, in addition to more straightforward soft tissues.

The raw metrics from the sequencing of the single nucleus libraries are presented in Table 7. The new protocol resulted in a substantial increase in the quality control metrics for both tissues. Compared to the previous protocol, the new one results in the sequencing of a significantly higher fraction of the transcripts contained in the library ("sequencing saturation")–an increase from 20%-25% to 53%-70%. This translates to capturing more mRNAs in the libraries with the

**Table 7. Sequencing metrics for two biological replicate libraries created with the protocol V1 [7] and two biological replicates for the protocol presented here (V2).**

|  | V1a | V1b | V2a | V2b |
|---|---|---|---|---|
| Number of reads | $2.30 \times 10^8$ | $2.29 \times 10^8$ | $2.60 \times 10^8$ | $2.29 \times 10^8$ |
| Uniquely mapped reads | 16.5% | 22.3% | 59.3% | 53.5% |
| Mapping to introns + exons | 7.81% | 11.2% | 50.2% | 45.1% |
| Sequencing saturation | 20.0% | 25.6% | 71.0% | 53.2% |
| Number of UMIs | 400,7192 | 5,577,109 | 10,055,941 | 11,059,639 |
| Number of nuclei | 4115 | 6018 | 2176 | 2832 |
| Median UMIs/nucleus | 767 | 774 | 3583 | 3088 |
| Median genes/nucleus | 618 | 653 | 2229 | 2042 |

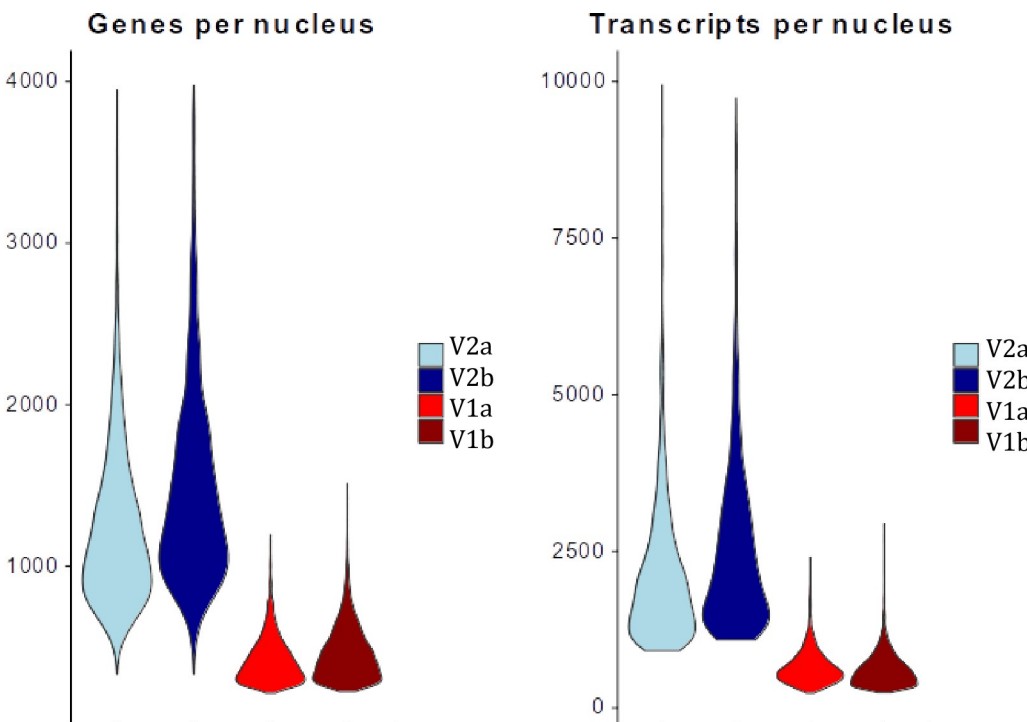

**Fig 2. Violin plots visualizing transcript and gene numbers per nucleus in each of the test datasets.** A comparison of transcript and genes numbers obtained using the older version of the TST protocol (V1, red) compared to the protocol in this paper (V2, blue). All libraries are generated from single nucleus suspensions isolated from Atlantic salmon skin samples, an extremely challenged tissue type to perform single nucleus sequencing.

same amount of sequencing. The new version of the protocol sees an increase in both the number of UMIs per nucleus (767–774 to 3088–3583) and in the number of genes per nucleus (618–653 to 2042–2229; Fig 2). This is the result of i) using RNA inhibitor to stop the degradation of RNA during nuclear isolation and library preparation, and ii) changing the loading buffer to PBS+BSA, since the original ST loading buffer has salts that can interfere with the microfluidic chemistry. This increase in data quality will result in improved downstream analyses such as the identification of cell types and performing of differential gene expression tests.

This new protocol has been successfully used to obtain nuclei from a variety of Atlantic salmon (*Salmo salar*) tissue types, including; liver, spleen, whole blood, leukocytes isolated from the blood, head kidney, gill, fin, as well as skin. It has also been used in our hands to successfully obtain nuclei from various tissues in other species, including the spleen of sheep (*Ovis aries*), chicken (*Gallus gallus*) mouse (*Mus musculus*) and nurse shark (*Ginglymostoma cirratum*), the olfactory organs of rabbit (*Oryctolagus cuniculus*) and Senegalese sole (*Solea senegalensis*), and pig (*Sus domesticus*) intestine. The protocol has also been used to obtain nuclei from invertebrate species including whiteleg shrimp (*Litopenaeus vannamei*) hepatopancreas and whole sea lice (*Lepeophtheirus salmonis*) at the copepod life stage. Based on these test cases, and the results presented here, we are confident the protocol will have wide future applications for snRNA-Seq in a variety of species and tissues.

## Supporting information

**S1 File.**
(PDF)

## Acknowledgments

### Ethics statement

The animal study was reviewed and approved by Centre for Aquaculture technologies and was carried out in compliance guidelines set forth by the Canadian Council for Animal Care, the animal use protocol was approved by the institutional Animal care committee MB-0185.

## Author Contributions

**Conceptualization:** Rose Ruiz Daniels, Ross Dobie, Daniel J. Macqueen, Diego Robledo.

**Data curation:** Rose Ruiz Daniels.

**Formal analysis:** Rose Ruiz Daniels, Richard S. Taylor, Diego Robledo.

**Investigation:** Rose Ruiz Daniels, Richard S. Taylor, Ross Dobie, Daniel J. Macqueen, Diego Robledo.

**Methodology:** Rose Ruiz Daniels, Richard S. Taylor, Sarah Salisbury, James J. Furniss, Emily Clark, Daniel J. Macqueen, Diego Robledo.

**Project administration:** Rose Ruiz Daniels, Richard S. Taylor, Diego Robledo.

**Resources:** Diego Robledo.

**Supervision:** Rose Ruiz Daniels, Daniel J. Macqueen, Diego Robledo.

**Validation:** Rose Ruiz Daniels, Daniel J. Macqueen.

**Visualization:** Rose Ruiz Daniels.

**Writing – original draft:** Rose Ruiz Daniels, Richard S. Taylor, Daniel J. Macqueen, Diego Robledo.

**Writing – review & editing:** Rose Ruiz Daniels, Richard S. Taylor, Daniel J. Macqueen, Diego Robledo.

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
