## [Decision Letter · Decision Letter 0]

28 Feb 2023

PONE-D-22-34938

A versatile nuclei extraction protocol for single nucleus sequencing in non-model species – optimization in various Atlantic salmon tissues

PLOS ONE

Dear Dr. daniels,

Thank you for submitting your manuscript to PLOS ONE. After careful consideration, we feel that it has merit but does not fully meet PLOS ONE’s publication criteria as it currently stands. Therefore, we invite you to submit a revised version of the manuscript that addresses the points raised during the review process.

We look forward to receiving your revised manuscript.

Kind regards,

Christopher W Reid, Ph.D

Academic Editor

PLOS ONE

Journal Requirements:

3. Please ensure that you include a title page within your main document. You should list all authors and all affiliations as per our author instructions and clearly indicate the corresponding author.

4. Please amend your manuscript to include your abstract after the title page.

7. We note you have not yet provided a protocols.io PDF version of your protocol and/or a protocols.io DOI. When you submit your revision, please provide a PDF version of your protocol as generated by protocols.io (the file will have the protocols.io logo in the upper right corner of the first page) as a Supporting Information file. The filename should be S1_file.pdf, and you should enter “S1 File” into the Description field. Any additional protocols should be numbered S2, S3, and so on. Please also follow the instructions for Supporting Information captions [https://journals.plos.org/plosone/s/supporting-information#loc-captions]. The title in the caption should read: “Step-by-step protocol, also available on protocols.io.”

Please assign your protocol a protocols.io DOI, if you have not already done so, and include the following line in the Materials and Methods section of your manuscript: “The protocol described in this peer-reviewed article is published on protocols.io (https://dx.doi.org/10.17504/protocols.io.[...]) and is included for printing purposes as S1 File.” You should also supply the DOI in the Protocols.io DOI field of the submission form when you submit your revision.

If you have not yet uploaded your protocol to protocols.io, you are invited to use the platform’s protocol entry service [https://www.protocols.io/we-enter-protocols] for doing so, at no charge. Through this service, the team at protocols.io will enter your protocol for you and format it in a way that takes advantage of the platform’s features. When submitting your protocol to the protocol entry service please include the customer code PLOS2022 in the Note field and indicate that your protocol is associated with a PLOS ONE Lab Protocol Submission. You should also include the title and manuscript number of your PLOS ONE submission.

Additional Editor Comments:

Please pay particular attention to Reviewer #1 comments regarding a comparable assay to the one described. I would agree that the method cited as the comparison is not ideal for comparing to the reported new technique for extracting nuclei from difficult samples. Before proceeding, I would like to see this properly addressed.

Reviewers' comments:

Reviewer's Responses to Questions

**Comments to the Author**

1. Does the manuscript report a protocol which is of utility to the research community and adds value to the published literature?

Reviewer #1: Yes

Reviewer #2: Yes

2. Has the protocol been described in sufficient detail?

To answer this question, please click the link to protocols.io in the Materials and Methods section of the manuscript (if a link has been provided) or consult the step-by-step protocol in the Supporting Information files.

The step-by-step protocol should contain sufficient detail for another researcher to be able to reproduce all experiments and analyses.

Reviewer #1: Yes

Reviewer #2: Yes

3. Does the protocol describe a validated method?

Reviewer #1: Yes

Reviewer #2: Yes

4. If the manuscript contains new data, have the authors made this data fully available?

Reviewer #1: Yes

Reviewer #2: No

**5. Is the article presented in an intelligible fashion and written in standard English?**

Reviewer #1: Yes

Reviewer #2: Yes

6. Review Comments to the Author

Reviewer #1: In this manuscript the authors describe a method of nuclei purification that increases yield and quality of the collected sample over previous methods. The authors also present data comparing the end result of their method to the previously used method. Overall, the steps of the manuscript are clear, and the protocol meets its claim as being better suited than the previous method.

Major issues:

Though the data is convincing, I’m not sure that the pervious method of nuclei purification was a truly suitable method for the types of tissues described here, ie. frozen and tough tissue samples. The citation given links to methods used for soft laboratory derived tissues. Considering that I wonder what the justification of using the previous approach would be. Searching the methodologies of projects purifiy nuclei from hard to access tissues, these also use mechanical methods (such as grinding and cutting the tissue). These methodologies however maybe more appropriate as points of comparison.

Reviewer #2: This lab protocol article gives a detailed and easy to follow description of a protocol for extracting nuclei from diverse tissue types and species, for single nucleus sequencing. The article is well written, and I think it provides enough detail for the protocol to be reproduced by other users with varied levels of lab experience. I think this is an excellent protocol to be published in PLOS one. The authors have applied this protocol to a wide range of species and tissue types already which I think demonstrates the potential for the protocol to be used extensively. The quality of single nucleus RNA sequencing data produced from the protocol establish that the protocol could see extensive use for investigating many topics.

I have a few minor comments and suggestions.

TST (4ml) reagent table: Should that be 1,840µl of nuclease-free water to make 4000?

Note at the top of page 10: Can you define nucleus clumping. How would the inexperienced user identify that this is happening?

Workflow step 2: “Keeping the culture place on ice” should that be “placed” on ice?

Workflow step 5: “Gently resuspend the pellet in PBS-BSA using a P1000 pipette” do you mean you remove the supernatant with the pipette and you don’t just pour it off before resuspending?

Results and discussion page 12: “This translates to capturing more mRNAs in the libraries with the same amount of sequencing.” At this point I think it is worth adding more of a discussion about the main reasons why you saw this improvement.

Results and discussion page 13: “The protocol has also also” remove second also

The data from the trail will also have to be uploaded to a public repository.

7. PLOS authors have the option to publish the peer review history of their article (what does this mean?). If published, this will include your full peer review and any attached files.

Reviewer #1: No

Reviewer #2: No

---

## [Author Response · Author response to Decision Letter 0]

27 Mar 2023

Dear editor and reviewers,

Many thanks for your constructive and helpful comments. These comments have really improved the clarity of the manuscript, and we want to thank the reviewers for their time and effort. We believe their comments have made this protocol more accessible. We have incorporated all the suggestions to the best of our ability. Please find below our point-by-point response. We hope you will find this revised version acceptable for publication. We look forward to hearing your comments.

Sincerely,

Rose Ruiz Daniels

Reviewer 1

In this manuscript the authors describe a method of nuclei purification that increases yield and quality of the collected sample over previous methods. The authors also present data comparing the end result of their method to the previously used method. Overall, the steps of the manuscript are clear, and the protocol meets its claim as being better suited than the previous method.

We thank the reviewer for the positive comments.

Major issues:

Though the data is convincing, I’m not sure that the pervious method of nuclei purification was a truly suitable method for the types of tissues described here, ie. frozen and tough tissue samples. The citation given links to methods used for soft laboratory derived tissues. Considering that I wonder what the justification of using the previous approach would be. Searching the methodologies of projects purifiy nuclei from hard to access tissues, these also use mechanical methods (such as grinding and cutting the tissue). These methodologies however maybe more appropriate as points of comparison.

Thank you, this is a very good point. We have addressed this in the introduction, where we state why ST with chopping is the most appropriate method for this sort of tissue (L111-L115). We have added the appropriate reference to justify the choice of this protocol, and cited previous work that tested these protocols in comparison with other known methods used to purify nuclei, including mechanical methods and gradients. These show that the method we chose for comparison was the best alternative, reducing background contamination while capturing cellular and transcript diversity.

Minor issues: 

The second paragraph of the introduction has a lot of “should” “could” “would” “anticipate”. Ideal the paragraph would read more like the first and be more direct and succinct. 

Agreed, we have changed the text according to the reviewer suggestions (L120-L130).

Using a glycerol cushion to remove debris instead of a filter. 

In order to not clog up the 10X genomics microfluidic device at least a 40 um filter is required – this is the manufacturers recommendation for the device. While a glycerol cushion may work, the protocol must be carried out at 4C, which would impact the viscosity of the glycerol and increase the protocol time - both of which may have negative consequences on nuclei viability.

Not sure why the whole protocol is rehashed in the “testing the protocol” section. In this section it seems that it would be most appropriate to explain what experiment was set up or just scrap it and have a results section.

We have improved this section by removing the repetition and instead describing the optimizations in the new protocol (L299-L311)

Citations were done improperly, please check the Plos One guidelines.

Thank you for pointing this out. We have corrected the references.

It is hard to determine what the actual changes in the protocol are actually doing and whether or not this group is just much better than others at collecting nuclei. Part of this I believe is the structure of the article. 

We have clarified the justifications for the modifications in L291-297. We have tested both versions of the protocol in-house, with and without the changes and the purpose of this section and results is to show the changes improved the protocol substantially. In particular, the extraction of nuclei out of very challenging tissues such as skin is improved. We present results that shows that these changes have had an effect on library quality.

Line 21 – typo, word missing

Corrected, thank you.

Line 131 – clarify what is adjusted. This statement about the time isn’t very clear. It might be clearer to state that dissociation takes up to 10 minutes. “First 5 minutes of mincing and then pipetting for up to 5 minutes checking every minute for completion of nuclei dissociation.” 

We have tried to clarify this (L236-L243) and the passage now reads: “The total time in the dissociation buffer is critical and the duration of the scissor and pipette steps needs optimisation using non-valuable samples. The use of scissors should be the minimum time necessary for no solid lumps of tissue to remain. Pipetting duration is optimised by observing the nuclei under a microscope and adjusting the dissociation time to the minimum needed for full dissociation and lysing of cells (Figure 1). Typical tissue dissociation takes a total of 5-10 minutes. Non-tissue samples such as blood samples require as little as 1 minute of gentle pipetting in the TST and no use of scissors.”

Line 132 – two sentences

Corrected.

Line 140 – to be more universal, should the centrifuge time just be set to 10 minutes.

Good point. The reason we do not feel 10 minutes is always appropriate is that longer duration (and higher speeds) can result in significant clumping of nuclei in samples which large recovery rates. We have clarified this point in the text – it should useful to readers as this is an issue we have encountered in the past: “When nuclei yield is low, centrifugation time can be increased to 10 minutes to maximise yield. In samples with high recovery 10 minutes is not recommended as longer centrifugation duration can result in the clumping of nuclei and higher doublet rates in the final data.”

Line 163 – seems that this should be stated in the discussion or at the least a sub step of step 7. This also seems to be a specific type of equipment that isn’t available to everyone, where as a haemocytometer is more readily available. 

Any haemocytometer that can be used for counting nuclei is suitable and text adjusted accordingly.

Line 180 – should be reworded to not “hang” on the citation or state “from Drokhlyansky and colleagues4”. There are several places in the manuscript this is done and there are ways around it. 

We have adjusted this in the text to improve readability.

Line 180-206 – It sounds like you tested your version to the previous published version. I believe that you should just state that and reference the changes you made. I’m not sure why the whole protocol is restated. 

Agreed, we have modified this section which now only states the changes (L291-297)

Figure 2 - Not sure why figure two isn’t just a bar graph. Using these shapes indicates that there is other information here. If that is the case, it should be explain, otherwise standard bar graphs should be used. 

The violin plots describe the distribution of transcripts and genes per nucleus across the whole dataset, which in heterogeneous tissue samples such as skin can be highly variable across cell types. For this reason, violin plots are commonly used in single cell studies e.g. a tissue sample with only two cell types with very different transcriptional activity would show a bimodal distribution in a violin plot, which is missed in a standard bar graph. We have changed the figure text to explain this: 

Figure 2 – typo in the legend

We have made edits to the legend: “Violin plots visualizing transcript and gene numbers per nucleus in each of the test datasets. A comparison of transcript and genes numbers obtained using the older version of the TST protocol (V1, red) compared to the protocol in this paper (V2, blue). All libraries are generated from single nucleus suspensions isolated from Atlantic salmon skin samples, an extremely challenged tissue type to perform single nucleus sequencing.”

Reviewer #2: This lab protocol article gives a detailed and easy to follow description of a protocol for extracting nuclei from diverse tissue types and species, for single nucleus sequencing. The article is well written, and I think it provides enough detail for the protocol to be reproduced by other users with varied levels of lab experience. I think this is an excellent protocol to be published in PLOS one. The authors have applied this protocol to a wide range of species and tissue types already which I think demonstrates the potential for the protocol to be used extensively. The quality of single nucleus RNA sequencing data produced from the protocol establish that the protocol could see extensive use for investigating many topics.

We thank the reviewer for their positive comments, we are really glad that they think this protocol could be widely useful to other researchers. We certainly hope so.

I have a few minor comments and suggestions.

TST (4ml) reagent table: Should that be 1,840µl of nuclease-free water to make 4000? 

Correct! We have amended this.

Note at the top of page 10: Can you define nucleus clumping. How would the inexperienced user identify that this is happening? 

We have amended this text, which now reads:

“It is possible that nuclei may stick together in clumps after centrifugation. This can be observed under a microscope (figure 1A illustrates an example of this). This is undesirable as it may lead to high rates of “doublets” (more than one nucleus being encapsulated in a single droplet in the microfluidics system, resulting in each transcript being labelled with the same cellular barcode).”

Workflow step 2: “Keeping the culture place on ice” should that be “placed” on ice?

Correct, “place” was a typo that now reads “plate”.

Workflow step 5: “Gently resuspend the pellet in PBS-BSA using a P1000 pipette” do you mean you remove the supernatant with the pipette and you don’t just pour it off before resuspending?

We have clarified the text to mentioned removing the supernatant prior to resuspension.

Results and discussion page 12: “This translates to capturing more mRNAs in the libraries with the same amount of sequencing.” At this point I think it is worth adding more of a discussion about the main reasons why you saw this improvement. 

The improved results are the combined effect of the use of RNAse inhibitor and the change in loading buffer. The impact of RNAse inhibitor is relatively straightforward, since it stops the degradation of RNA during the nuclear isolation and library preparation stages. Regarding loading buffer, the previous one (ST) had salts that can interfere with the microfluidic chemistry. We have added this information as suggested (L350-353).

Results and discussion page 13: “The protocol has also also” remove second also

Corrected, thank you.

The data from the trail will also have to be uploaded to a public repository.

We have now uploaded the data to a public repository. We have yet to obtain an accession from NCBI but this will be added prior to publication. We have added a data availability statement (L368-L371) with a placeholder for the accession number.

---

## [Decision Letter · Decision Letter 1]

13 Apr 2023

A versatile nuclei extraction protocol for single nucleus sequencing in non-model species – optimization in various Atlantic salmon tissues

PONE-D-22-34938R1

Dear Dr. Ruiz Daniels,

We’re pleased to inform you that your manuscript has been judged scientifically suitable for publication and will be formally accepted for publication once it meets all outstanding technical requirements.

Kind regards,

Sven Winter

Academic Editor

PLOS ONE

Additional Editor Comments (optional):

Reviewers' comments:

Reviewer's Responses to Questions

**Comments to the Author**

1. Does the manuscript report a protocol which is of utility to the research community and adds value to the published literature?

Reviewer #1: Yes

Reviewer #2: Yes

2. Has the protocol been described in sufficient detail?

To answer this question, please click the link to protocols.io in the Materials and Methods section of the manuscript (if a link has been provided) or consult the step-by-step protocol in the Supporting Information files.

The step-by-step protocol should contain sufficient detail for another researcher to be able to reproduce all experiments and analyses.

Reviewer #1: Yes

Reviewer #2: Yes

3. Does the protocol describe a validated method?

Reviewer #1: Yes

Reviewer #2: Yes

4. If the manuscript contains new data, have the authors made this data fully available?

Reviewer #1: Yes

Reviewer #2: Yes

**5. Is the article presented in an intelligible fashion and written in standard English?**

Reviewer #1: Yes

Reviewer #2: Yes

6. Review Comments to the Author

Reviewer #1: To the Editor,

I believe that the changes made by the authors of the Ruiz Daniels manuscript are sufficient and would support the decision of the editor to accept this paper.

Sincerely,

Steven Weicksel

Reviewer #2: This is the second time I have reviewed this manuscript and I think the reviewers have done an excellent job of responding to all reviewer comments. I also think this clearly described protocol could be useful to many projects in the future.

7. PLOS authors have the option to publish the peer review history of their article (what does this mean?). If published, this will include your full peer review and any attached files.

Reviewer #1: No

Reviewer #2: **Yes: **Samuel C. Andrew

---

## [Editor Report · Acceptance letter]

11 May 2023

PONE-D-22-34938R1 

A versatile nuclei extraction protocol for single nucleus sequencing in non-model species – optimization in various Atlantic salmon tissues 

Dear Dr. Ruiz Daniels:

I'm pleased to inform you that your manuscript has been deemed suitable for publication in PLOS ONE. Congratulations! Your manuscript is now with our production department. 

Kind regards, 

on behalf of

Dr. Sven Winter 

Academic Editor

PLOS ONE